# Microbial Diversity in Subarctic Biocrusts from West Iceland following an Elevation Gradient

**DOI:** 10.3390/microorganisms9112195

**Published:** 2021-10-21

**Authors:** Ekaterina Pushkareva, Israel Barrantes, Peter Leinweber, Ulf Karsten

**Affiliations:** 1Department of Applied Ecology and Phycology, Institute of Biological Sciences, University of Rostock, 18059 Rostock, Germany; ulf.karsten@uni-rostock.de; 2Department of Biology, Botanical Institute, University of Cologne, 50674 Cologne, Germany; 3Research Group Translational Bioinformatics, Institute for Biostatistics and Informatics in Medicine and Ageing Research, Rostock University Medical Center, 18057 Rostock, Germany; israel.barrantes@uni-rostock.de; 4Department Soil Sciences, Faculty of Agricultural and Environmental Sciences, University of Rostock, Justus-von-Liebig-Weg 6, 18051 Rostock, Germany; peter.leinweber@uni-rostock.de

**Keywords:** Iceland, biocrust, diversity, bacteria, cyanobacteria, fungi, eukaryotes, high-throughput sequencing, culturing

## Abstract

Biological soil crusts (biocrusts) are essential communities of organisms in the Icelandic soil ecosystem, as they prevent erosion and cryoturbation and provide nutrients to vascular plants. However, biocrust microbial composition in Iceland remains understudied. To address this gap in knowledge, we applied high-throughput sequencing to study microbial community composition in biocrusts collected along an elevation gradient (11–157 m a.s.l.) stretching away perpendicular to the marine coast. Four groups of organisms were targeted: bacteria and cyanobacteria (16S rRNA gene), fungi (transcribed spacer region), and other eukaryotes (18S rRNA gene). The amplicon sequencing of the 16S rRNA gene revealed the dominance of Proteobacteria, Bacteroidetes, and Actinobacteria. Within the cyanobacteria, filamentous forms from the orders Synechococcales and Oscillatoriales prevailed. Furthermore, fungi in the biocrusts were dominated by Ascomycota, while the majority of reads obtained from sequencing of the 18S rRNA gene belonged to Archaeplastida. In addition, microbial photoautotrophs isolated from the biocrusts were assigned to the cyanobacterial genera *Phormidesmis*, *Microcoleus*, *Wilmottia*, and *Oscillatoria* and to two microalgal phyla Chlorophyta and Charophyta. In general, the taxonomic diversity of microorganisms in the biocrusts increased following the elevation gradient and community composition differed among the sites, suggesting that microclimatic and soil parameters might shape biocrust microbiota.

## 1. Introduction

Biological soil crusts (biocrusts) are assemblages of organisms, such as bacteria, cyanobacteria, fungi, eukaryotic microalgae, lichens, and mosses, living on the soil surface. They can survive long periods of droughts and live under extreme conditions with strong temperature and light gradients. Biocrust organisms are poikilohydric and, thus, can be rapidly reactivated once water is available. Moreover, biocrusts play an important role in primary production, nutrient cycling, mineralization, weathering, and soil stabilization [1]. Microbial carbon dioxide (CO_2_) fixation is mostly provided by photoautotrophic microorganisms, which are prominent members of biocrusts. Cyanobacteria, for example, usually participate in biogeochemical cycles such as carbon (C) or nitrogen (N) fixation [2]. Furthermore, cyanobacteria promote biocrust development forming a cohesive layer with soil particles, which increases surface stability and inhibits erosion [3]. Non-photoautotrophic bacterial communities in biocrusts typically consist of a number of ubiquitous phyla including Actinobacteria, Bacteroidetes, Proteobacteria, and Verrucomicrobia [4,5]. Proteobacteria and Actinobacteria that typically exhibit wide metabolic ranges, competitive advantages, and multiple UV repair mechanisms are functionally important in nutrient-limited arid environments [6]. Eukaryotic microbial members of biocrust communities are usually represented by fungi and microalgae. Fungi, including Basidiomycota and Ascomycota, are ecologically important, as they can release bioavailable nutrients and form symbiotic associations with microbial photoautotrophs into lichens [7]. In general, fungi produce vegetative filaments (hyphae) that bind soil particles together and help to consolidate the surface of biocrust communities. Eukaryotic microalgae, similar to cyanobacteria, contribute to the C cycle through their photosynthesis. Due to the structural and functional differences in photosynthetic organelles, eukaryotic microalgae have higher rates of photosynthesis than cyanobacteria but a lower tolerance toward freeze–thaw cycles [8]. While most knowledge about biocrust community composition and ecological functions originates from drylands, much less is known from higher latitudes [9].

Iceland, located right below the Arctic Circle, is a geologically young island with frequent volcanic eruptions. Icelandic soils, known as Andosols, are derived from volcanic ash and, therefore, are very distinct from other soils around the world [10]. They are susceptible to weathering which, in turn, results in the release of aluminum (Al), iron (Fe), and silicon (Si) [11]. Moreover, thin Andosols together with a harsh climate slow down vascular plant colonization in Iceland and, hence, biocrusts occupy considerable areas located on free spaces between the higher plants [12].

Despite the importance of biocrusts in high latitude areas, particularly Iceland, there are a limited number of studies about their microbial community composition. Most of the available literature focuses on the microbial communities in the Icelandic soil environment in general. For example, bacteria were characterized in volcanic deposits in Northern Iceland using high-throughput sequencing (HTS) which revealed the dominance of Proteobacteria, Acidobacteria, Actinobacteria, and Firmicutes [13]. Likewise, HTS was applied to study bacterial diversity in the deep soil (below 145 m depth) of the volcanic island Surtsey, and the majority of reads belonged to Methanobacteriales and Archaeoglobales [14]. Furthermore, arbuscular mycorrhizal fungi [15,16] and other terrestrial fungi [17,18] were described in Icelandic soils using morphological methods. Similarly, the diversity of cyanobacteria and eukaryotic microalgae was only reported in papers from the last century [19,20]. However, our recent study [12] based on a morphological approach revealed unexpectedly rich communities of microbial photoautotrophs in Icelandic biocrusts. Eukaryotic microalgae, such as green algae (*Chlorella*, *Coccomyxa*, and *Stichococcus*) and diatoms (e.g., *Pinnularia* and *Eunotia*) dominated these biocrusts [12].

Given the lack of information on microbial community composition inhabiting Icelandic biocrusts, the objective of this study was to obtain an overview of bacteria, cyanobacteria, fungi, and other eukaryotes in the biocrusts of west Iceland using HTS. Moreover, we hypothesized that different elevations (from 11 to 156 m a.s.l.) and distance from the sea (from 0 to 80 km) as well as soil properties would affect microbiota in these biocrusts.

## 2. Materials and Methods

### 2.1. Site Description

Litla-Skard (64°43’36” N, 21°37’48” W) is a research station in the west of Iceland approximately 100 km farther north from Reykjavik and belongs to the National Forestry Service (Icelandic Forest Service). The area has a subarctic climate with an annual average temperature of 3.1 °C. The coldest and warmest months are February and July with mean temperatures of −1.8 and 10.8 °C, respectively. The average annual precipitation is 930 mm in the form of rain and 100 mm in the form of snow (meteorological data are provided from the research station Litla-Skard; no online data are available). The vegetation is represented by shrub birches, moss heaths, marsh grass, and grassland. The soils in this area are Andosols, which have coarse texture, high porosity, low cohesion, and the dispersal of many layers of tephra caused by frequent volcanic eruptions.

Five sampling sites were chosen for this study that represented a catena of approximately 80 km in distance from the sea into inland and were located at different elevations from 11 to 157 m a.s.l.: Krákunes (KRA; 11 m a.s.l., at the sea coast), Borgarfjarðarbraut (BOR; 25 m a.s.l., 60 km from the sea), Fiflholt (FIF; 46 m a.s.l., 11 km from the sea), Litla-Skard station (LSK; 106 m a.s.l., 50 km from the sea), and Giljar (GIL; 157 m a.s.l., 80 km from the sea).

### 2.2. Sampling and Soil Characterization

Sampling was conducted in July 2014. Biocrust samples (up to 1 cm in depth) were taken in 4 replicates from each site and kept moist and cool until they were delivered to the laboratory where they were stored in a freezer.

The description of the samples (soil chemistry, sequential phosphorus fractionation, microbial P, and potential activity of phosphatase as well as community composition of algae and fungi assessed by microscope) are reported in Pushkareva et al. [12]. In brief, the biocrusts were acidic (pH 5.5–5.8) with a high P content (982–1571 mg kg^−1^) and moderately labile P was dominant. Microphotoautotrophic communities were dominated by eukaryotic microalgae including diatoms. Green algae, such as *Chlorella vulgaris*, *Chlamydomonas* sp., and *Coccomyxa* sp., and diatoms, such as *Pinnularia* sp. and *Eunotia* sp., were present in all sites. The majority of cyanobacteria belonged to filamentous forms such as *Microcoleus* sp., *Leptolyngbya* sp., *Phormidesmis* sp., and *Stenomitos* sp. In addition, the majority of fungal strains isolated from the biocrusts were assigned to *Penicillium* sp. and *Fusarium* sp. [12]. Moreover, mosses and lichens were not observed in the samples.

### 2.3. Cultivation and Molecular Analyses of Microbial Photoautotrophs

Cyanobacteria and eukaryotic microalgae were isolated from the enrichment cultures used in our previous study [12]. Single colonies were transferred to the Petri dishes with Bold’s Basal Medium, and unialgal cultures were established. The cultures were then kept at 15 °C under 30 μmol photons m^−2^ s^−1^ (Osram Lumilux Cool White lamps L36W/840) with a light/dark regime of 16/8 h. 

The DNA of cyanobacterial and microalgal strains was extracted using the NucleoSpin Plant II mini kit (Macherey Nagel, Düren, Germany) according to the manufacturer’s instructions. The 16S (for cyanobacteria) and 18S rRNA (for eukaryotic microalgae) genes were amplified using primers BS-1F/CPL-10R and EAF3/ITS055R, respectively (Appendix A). PCRs for both sets of primers were performed as follows: an initial denaturation step at 95 °C for 3 min, followed by 35 cycles of DNA denaturation at 95 °C for 15 s, primer annealing at 55 °C for 15 s, strand extension at 72 °C for 1 min 30 s, and a final extension step at 72 °C for 2 min. Sanger sequencing was performed at GATC Sequencing Services (Eurofins Genomics Germany, Ebersberg, Germany) with the primers listed in the Appendix A.

### 2.4. Environmental DNA Extraction and Sequencing

Total DNA was extracted from the biocrust samples using the PowerSoil DNA Isolation Kit (MOBIO, Carlsbad, CA, USA) according to the manufacturer’s instructions. DNA concentrations were quantified using a Qubit3.0 Fluorometer according to the manufacturer’s protocol. Extracted DNAs were sent to the Microsynth AG (Balgach, Switzerland), where PCR and sequencing using the Illumina MiSeq platform were performed.

Different sets of primers were used to target four groups of organisms: bacteria, cyanobacteria, fungi, and other eukaryotes (Appendix A). Universal bacterial primers (341F and 802R) and cyanobacteria specific primers (CYA_F and CYA_R) were used to amplify the V3–V4 region of the 16S rRNA gene, while primers tarEuk_F and tarEuk_F were used for the 18S rRNA gene amplification. To assess fungal community composition, the internal transcribed spacer region (ITS) was amplified with primers Fungi_F and Fungi_R. The raw reads were submitted to the European Nucleotide Archive (ENA) under the project PRJEB45587.

### 2.5. Bioinformatic and Statistical Analyses

Sequence contigs of isolates were generated using the software Geneious 8.1.9 (Biomatters Ltd., Auckland, New Zealand) and then deposited in GenBank with accession numbers MZ020203–MZ020221 for eukaryotic microalgae and MZ020189–MZ020202 for cyanobacteria. The sequences of the most closely related isolates from GenBank were retrieved using BLAST.

The amplicons were reconstructed from the Illumina sequencing runs using the *pandaseq* program (version 2.11; [21]). Operational taxonomic units (OTUs) were then identified from these amplicons using USEARCH (version 6.1.544; [22]) with the *pick_open_reference_otus.py* script of QIIME 1.9.1 [23]. The following databases were used: 16S rRNA Greengenes (version 13.8; [24]) for the bacteria and cyanobacteria; UNITE (version 12_11; [25]) for the ITS amplicons; and SILVA (version 132; [26]) for the 18S rRNA data. In all cases, a cutoff of 97% identity was applied. Afterwards, low confidence OTUs were excluded via the *remove_low_confidence_otus.py* script [27]. OTUs classified as chloroplasts and mitochondria were removed from the bacterial data set, while OTUs assigned to bacteria were deleted from the 18S rRNA results. Likewise, non-cyanobacterial OTUs were removed from the data set obtained with cyanobacteria-specific primers with additional manual inspection.

One replicate of site KRA (KRA-2) revealed only 31 reads with cyanobacteria-specific primers and, thus, was removed from further analyses. Likewise, this sample did not produce any reads using primers for 18S rRNA amplification. Moreover, two replicates were not included to alpha and beta diversity analyses of eukaryotes (18S rRNA data set) due to the low number of reads (LSK-2 and GIL-1 with 255 and 814 total reads, respectively).

Alpha diversity indices (OTU richness and Shannon’s diversity) were calculated in R (version 1.3.1073) using the package *vegan* [28]. The differences in parameters among sampling sites were tested with one-way analysis of variance (ANOVA) and Tukey’s HSD post hoc test using the software JMP 13.0.1 (SAS Institute Inc., Cary, NC, USA). A *p*-value < 0.05 was considered significant in all statistical tests. Normality of variance was assessed using the Shapiro–Wilk’s test. If necessary, data were Log or Box–Cox transformed. The relationships between diversity indices and soil parameters were tested by Pearson’s correlation coefficient (PCC). Community dissimilarities among the sites based on number of reads obtained by amplicon sequencing were assessed by non-metrical multidimensional scaling (NMDS) using the package *vegan* in R. In addition, permutational multivariate analysis of variance (PERMANOVA) was performed, and homogeneity of variances was tested by the betadisper command prior this analysis. Various taxonomic levels were used for both multivariate analyses: bacterial phyla, cyanobacterial orders, fungal phyla, and eukaryotic clades. Furthermore, Venn diagrams for each group of organisms were created using an online platform (http://bioinformatics.psb.ugent.be/webtools/Venn, accessed on 15 November 2020).

In addition, using FUNGuild database, fungal OTUs were assigned to functional (trophic) groups according to their nutrient acquisition strategy of the fungus: pathotrophs, saprotrophs, symbiotrophs, or a combination of these [29]. Furthermore, a PICRUSt analysis was carried out to predict the relative abundance of functional genes and pathways in the communities (version 2.11; [30]) for bacteria and cyanobacteria. For this, the OTUs were re-assigned with a closed-reference OTU picking protocol (from QIIME 1.9.1) and the Greengenes database (version 13.5), preclustered at 97% identity. The obtained OTU tables were then normalized by a 16S rRNA copy number, and functional genes were predicted from the Kyoto Encyclopedia of Genes and Genomes (KEGG) database [31]. Finally, the results from the PICRUSt analysis and KEGG predictions were processed within STAMP (version 2.1.3; [32]).

## 3. Results 

### 3.1. Overall Diversity of Microorganisms in the Icelandic Biocrusts

#### 3.1.1. Bacteria

Next generation sequencing with general bacterial primers resulted in 369,016 bacterial reads for 20 samples, which were clustered into 3155 OTUs. Of these, the majority of reads and OTUs belonged to the orders Proteobacteria (38% of total reads, 999 OTUs).

Bacteroidetes (15% of total reads, 368 OTUs), Actinobacteria (12% of total reads, 395 OTUs), Planctomycetes (9% of total reads, 396 OTUs), Acidobacteria (7% of total reads, 299 OTUs), Verrucomicrobia (6% of total reads, 242 OTUs), Chloroflexi (4% of total reads, 150 OTUs), Gemmatimonadetes (3% of total reads, 75 OTUs), and Cyanobacteria (1% of total reads, 42 OTUs) (Figure 1a). Within Proteobacteria, Alphaproteobacteria was the dominant class and constituted 18% of total bacterial reads (475 OTUs).

In addition, community function analysis revealed only one pathway in the 16S data set which significantly (*p* < 0.05) differed among the studied sites (Appendix A). A total of 74 bacterial genera were related to the glutathione metabolism pathway (K00480), and the majority of sequences assigned to it were recorded at site BOR.

#### 3.1.2. Cyanobacteria

Sequencing of the 16S rRNA gene with cyanobacteria-specific primers produced 835,441 reads (603 OTUs). After removal of non-cyanobacterial OTUs (other bacteria and chloroplasts), 244 OTUs (527,772 reads) assigned to cyanobacterial taxa remained. Six orders were distinguished according to the Greengenes database: Synechococcales (63% of total reads, 129 OTUs), Oscillatoriales (20% of total reads, 37 OTUs), Nostocales (13% of total reads, 53 OTUs), Chroococcales (2% of total reads, 12 OTUs), Gloeobacterales (<1% of total reads, 5 OTUs), and Stogonematales (<1% of total reads, 1 OTU) (Figure 1b). Seven OTUs (1% of total reads) could not be assigned to any order.

Ten strains of filamentous cyanobacteria isolated from the biocrusts were successfully sequenced (Table 1a). Sequences of six isolates were assigned to the order Oscillatoriales, particularly to the genera *Microcoleus* and *Wilmottia* (>94% identity). Further, sequences of four other isolates corresponded to *Phormidesmis* sp. from the order Synechococcales (>93% identity).

#### 3.1.3. Fungi

A total of 364,233 reads and 624 OTUs assigned to Fungi were obtained by sequencing the ITS region. Five fungal phyla were identified: Ascomycota (37% of total reads, 294 OTUs), Basidiomycota (17% of total reads, 89 OTUs), Zygomycota (7% of total reads, 10 OTUs), Glomeromycota (<1% of total reads, 3 OTUs), and Chytridiomycota (<1% of total reads, 2 OTUs) (Figure 1c). However, a large number of OTUs (38% of total reads, 226 OTUs) could not be assigned to any phyla. The majority of identified Ascomycota belonged to the classes Leotiomycetes, Sordariomycetes, and Eurotiomycetes (8%, 6%, and 5% of total fungal reads, respectively). Sequences affiliated with the class Agaricomycetes comprised the majority of those from Basidiomycota with 11% of total ITS reads.

Furthermore, 67% of fungal OTUs were identified by the FUNGuild database, and only 22% of identified OTUs were assigned to different functional groups. The majority of fungal reads belonged to symbiotrophs and saprotrophs (Figure 2).

In addition, fungal OTUs (132 OTUs) obtained by amplicon sequencing of the 18S rRNA constituted 10% of total eukaryotic reads and 58% of Opisthokonta clade. Phyla Mucoromycota and Cryptomycota exhibited the majority of reads within fungi (38 and 30% of total fungal reads, respectively).

#### 3.1.4. Eukaryotes

Sequencing of the 18S rRNA gene produced 850,183 reads and 1033 OTUs assigned to Eukaryota. The majority of reads affiliated with the clade Archaeplastida (57% of total reads, 325 OTUs) followed by clades SAR (23% of total reads, 425 OTUs), Opisthokonta (17% of total reads, 224 OTUs), Amoebozoa (3% of total reads, 56 OTUs), Centrohelida (<1% of total reads, 56 OTUs) and Excavata (<1% of total reads, 2 OTUs) (Figure 1d). Archaeplastida was represented by the clade Chloroplastida only which, in turn, included two algal phyla, Charophyta (80 OTUs) and Chlorophyta (245 OTUs). The SAR groups were almost evenly distributed within the clade: 30% (105 OTUs) affiliated with Stramenopiles, 34 % (109 OTUs) with Alveolata, and 36% (211 OTUs) with Rhizaria. The majority of reads within Stramenopiles were assigned to Bacillariophyceae (33% of total Stramenopiles reads; 31 OTUs).

In addition, 20 strains of eukaryotic microalgae were isolated and sequenced as previously described (Table 1b). Of these, the majority of the strains were from the order Chlorophyta, and only one strain isolated from site FIF affiliated with the order Charophyta (class Klebsormidiophyceae). Sites KRA and GIL had more distinct isolates in comparison to other sites and each other. For example, *Gonium* sp. was isolated only from site KRA (KRA-4a), while *Stichococcus* sp. (GIL-3a), *Nannochloris* sp. (GIL-3d), and *Parietochloris* sp. (GIL-4a) only from site GIL. The genera *Xerochlorella* and *Chlorella* were common for sites BOR, FIF, and LSK (Table 1b).

### 3.2. Alpha and Beta Diversities

In general, two diversity indices (OTU richness and Shannon index) showed a trend to increase following the elevation gradient from the sea, with some exceptions (Figure 3). A significant difference between the lowest and highest topographic elevations was observed in the diversity indices for bacteria, cyanobacteria, and fungi. However, a significantly higher OTU richness of eukaryotes and fungi was observed at site FIF. For the cyanobacteria, site BOR had significantly lower OTU richness and Shannon index than other sites. Further, community structure at the BOR site appeared distinct from those at the other sites (Figure 1). For example, significantly more reads from BOR affiliated with Proteobacteria, cyanobacterial order Synechococcales, and clade Archaeplastida than in biocrusts from the other sites. Similarly, most reads from the BOR site did not affiliate with a fungal phylum, and the highest relative abundance of pathotrophs was found here (Figure 2). In addition, Venn diagrams showed that more OTUs were shared among sites KRA, FIF, LSK, and GIL rather than with site BOR (Appendix A).

The NMDS plot revealed that microbial communities were distinct among the five sites and overall grouped by the site of origin, except site KRA, which exhibited a rather heterogeneous community composition (Figure 4). This was confirmed by PERMANOVA analysis (Table 2 and Appendix A). Moreover, community dissimilarities for each microbial group were explained by different environmental parameters, measured in our previous publication [12]. For example, significant correlations (*p* < 0.05) were observed between manganese (Mn), Fe, and Al contents and fungal community composition, while dissimilarities in the composition of bacterial communities were explained by C, N, and S contents, C:N ratio, and the potential activity of acid phosphatase. Furthermore, differences in cyanobacterial and eukaryotic community composition were related to Na and microbial P (only for eukaryotes) contents. In addition, Pearson’s correlation test revealed no significant correlations between alpha diversity indices and soil chemistry parameters.

## 4. Discussion

### 4.1. Overall Diversity of Microbiota in Icelandic Biocrusts

Most available studies about microbial biodiversity in biocrusts of the Northern Hemisphere are from Arctic regions (e.g., [33,34]). These investigations often reveal a high diversity of microorganisms in polar biocrusts and indicate the striking ecological importance of their microbiota in such high latitude areas. The subarctic zone, located right below the Arctic Circle, has received far less attention. Overall, the bacterial community in the studied Icelandic biocrusts comprised largely Proteobacteria, Bacteroidetes, and Actinobacteria (Figure 1a), which is consistent with previous observations from Arctic and other subarctic environments [35]. These bacteria play an important role in biocrust formation and maintenance due to the fact of their production of exopolysaccharides (Bacteroidetes), pigments (Proteobacteria), or mycelia (Actinobacteria) [36]. They are often observed in permafrost habitats and might be well adapted to cold conditions [37,38]. Moreover, soils with high organic matter content promote growth of Proteobacteria [39]. Andosols, derived from volcanic parent material, accumulate more organic matter than other soil types [40]. Other important bacteria, Planctomycetes, observed in the Icelandic biocrusts and also previously in Arctic soils, strongly participate in the nitrogen cycle by performing an anaerobic ammonium oxidation [35] and, therefore, might play an important role in biocrusts particularly from cold environments.

The assessment of cyanobacterial community composition using amplicon sequencing showed that, in general, the distribution of cyanobacterial orders in Icelandic biocrusts was similar to those from Svalbard archipelago using the same set of primers (Figure 1b; [41,42]). The biocrusts were dominated by filamentous forms from the order Synechococcales, and that is also in agreement with microscopic observations [12] and culture isolation (Table 1a). Moreover, *Leptolyngbya* sp., which was the major representative of the Synechococcales, is common in biocrusts of cold environments [3]. Interestingly, the strains of filamentous cyanobacteria from this order isolated from the Icelandic biocrusts were assigned to *Phormidesmis* sp. (>97% identity) and not *Leptolyngbya* sp. (Table 1a). This might indicate that some sequences stored in the database used for the identification of the reads obtained by amplicon sequencing have incorrect taxonomic names on the genus level. In addition, most of these strains were closely related to *Phormidesmis arctica* reported previously from Central Svalbard [43] and, furthermore, corresponded to the higher elevations of the Icelandic and Svalbard sampling sites. For example, a strain isolated from the Icelandic site GIL (GIL-3b) was similar to *Phormidesmis arctica* recorded in Mumien Peak (Svalbard) at the altitude of 442 m a.s.l. and strains from Icelandic sites BOR (BOR-4c) and FIF (FIF-1a) corresponded to one strain isolated from a glacier foreland at the lower altitude of Svalbard [43]. Such data reflect a high degree of genetic relatedness and, hence, it is possible to speculate that, for example, migrating birds, such as Arctic terns (*Sterna paradisea*), could act as a vector for transportation of cyanobacteria between Iceland and Svalbard. A considerable number of reads in the biocrusts obtained with amplicon sequencing were assigned to another filamentous order of cyanobacteria (Oscillatoriales), and the majority of cultured isolates belonged indeed to the Oscillatoriales. Similar to Synechococcales, members of this order can survive extreme conditions due to the array of adaptive strategies such as dormancy, motility, formation of associations, and production of mucilage [44]. 

The dominance of the fungal phyla Ascomycota, Basidiomycota, and Zygomycota has been described in different Arctic soils [35], and it was also observed in this study (Figure 1c). Several members of these phyla have evolved ecophysiological traits that permit their survival in cold environments such as production of antifreeze proteins (AFPs) as well as protective sugars, polyols, lipids, and fatty acids [45]. In addition, the high number of OTUs that could not be assigned to any phyla might indicate yet undescribed fungal species in Icelandic biocrusts. Therefore, more studies focusing on fungal isolation and following morphological and molecular determinations are required to confirm this fact. The widely used fungal database UNITE often identifies fungal sequences to a kingdom level only [46], which might also lead to a high number of non-assigned OTUs. In addition, higher relative abundance of saprotrophs occurred in the sites located further from the sea (i.e., BOR, LSK and GIL), which could be related to the higher amount of litter in the inland. These fungi play an important role in decomposition and stabilization of soil organic matter [47].

Archaeplastida, represented here only by Chlorophyta and Charophyta, numerically dominated the eukaryotic communities in terms of the number of sequences in each biocrust (Figure 1d). In agreement with the amplicon sequencing, the strains isolated from the biocrust samples belonged to these two phyla (Table 1b), which are very common in northern latitudes [3]. Moreover, all sequences of cultured isolates were found in the amplicon sequencing data set but were underrepresented in number, pointing to a large part of the microalgae remaining uncultivable under the conditions we used in this study. Diversity and abundance of green algae in polar biocrusts usually exceed those of yellow-green algae [48], which seems to be similar in Icelandic biocrusts. In addition, only one isolate and three OTUs of *Klebsormidium* sp. were reported in the Icelandic biocrusts. Despite their ability to live in cold environments due to the increased desiccation tolerance [49], they were also detected in low abundance in Svalbard biocrusts [4]. Diatoms constituted only 2% of total eukaryotic reads, which did not correspond to the results of microscopic observations that showed their dominance in the biocrusts. This could be due to the low primer coverage efficiency towards different eukaryotic groups. To the best of our knowledge, there are no primers available with 100% coverage of all eukaryotic microorganisms. In addition, the phylum Cercozoa (Rhizaria), a typical soil protist group, was the dominant representative of the SAR clade in the biocrusts. These single-celled eukaryotes have already been described here, with members of the order Glissomonadida dominating [50].

### 4.2. Shift of Microbial Community Composition in Biocrust Depending on Environmental Parameters

The microbial diversity in Icelandic biocrusts increased from the lowest to the highest topographic elevations (Figure 3), and several previous studies reached similar observations [42,51,52]. Perhaps, there is more water available in the soil at higher altitudes, which would promote microbial growth [53]. Furthermore, the samples grouping according to the site of origin (except site KRA) showed overall community homogeneity at each elevation (Figure 4). Site KRA located at the coast, however, exhibited a more heterogeneous community composition in comparison with all other sites, and this is in agreement with microscopic observations of microbial photoautotrophs [12]. Moreover, there was a lower bacterial diversity than at the other sites, which could be explained by frequent disturbance induced by marine water or sea spray. This supports a previous finding that salt in the coastal area negatively impacted soil microbial community composition [52]. Moreover, an increase of Bacteroidetes at site KRA is consistent with previous studies showing a positive correlation between electrical conductivity and abundance of Bacteroidetes in soils [54]. Likewise, the increase in the fungi Ascomycota at site KRA is similar to a previous study from India indicating a higher number of this phylum in sand dunes than in other soil environments [55]. In addition, sequences of two microalgal strains isolated from the site KRA shared >99% sequence identity with *Chlamydomonas hedleyi*, a typical marine species, and *Gonium pectorale*, a freshwater species (Table 1a). The latter is usually found in nutrient-rich lakes or ponds [56] but to the best of our knowledge has never been described in biocrusts.

Interestingly, community composition at site BOR, located further from the sea but at lower elevation, was distinct from other sites (Figure 1). This site showed a higher abundance of typical biocrust representatives such as Proteobacteria and filamentous forms of Synechococcales [33,41,52] than the other sites. Furthermore, the prevalent clade Archaeplastida was dominated by algae from the phylum Charophyta at site BOR unlike the other sites, where members of the phylum Chlorophyta prevailed. Perhaps, distance from the sea and low altitude played a role in community shaping rather than the chemical composition, as there were no prominent statistical correlations between alpha diversity indices and soil chemistry data. Nevertheless, there were positive correlations between two prokaryotic groups and fungi.

In addition, a glutathione metabolism pathway was revealed in several bacterial genera, and most of them were recorded at site BOR (Appendix A). Glutathione is involved in bacterial redox-regulation and acclimation to various stressors [57], which could explain higher proportions of sequences responsible for glutathione metabolism at BOR, with the lowest carbon and nitrogen contents and C/N ratio [12].

## 5. Conclusions

In this study, we showed that the biocrusts from west Iceland harbor diverse communities of various prokaryotic and eukaryotic microorganisms that differ in their composition depending on their microenvironment. Furthermore, we observed an increase in microbial diversity with increasing elevation.

This work provides a biodiversity baseline for understanding the structure and functions of biocrusts in this unique subarctic environment.

## Figures and Tables

**Figure 1 microorganisms-09-02195-f001:**
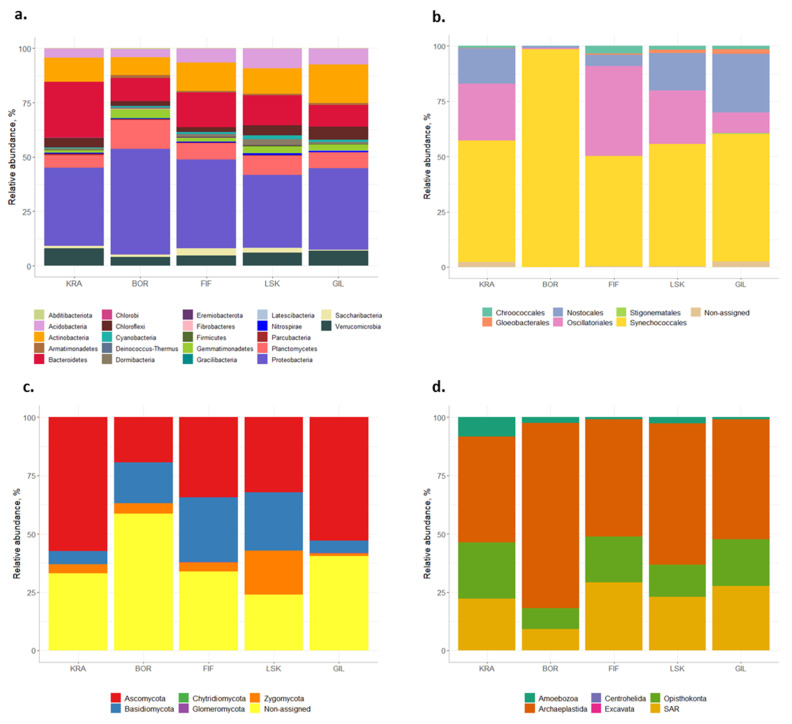
General microbial profile of Icelandic biocrusts. The figures show the relative abundances based on the read numbers obtained by amplicon sequencing: (**a**) bacterial phyla; (**b**) cyanobacterial orders, (**c**) fungal phyla, and (**d**) eukaryotic clades. KRA—Krákunes, BOR—Borgarfjarðarbraut, FIF—Fiflholt, LSK—Litla-Skard, GIL—Giljar.

**Figure 2 microorganisms-09-02195-f002:**
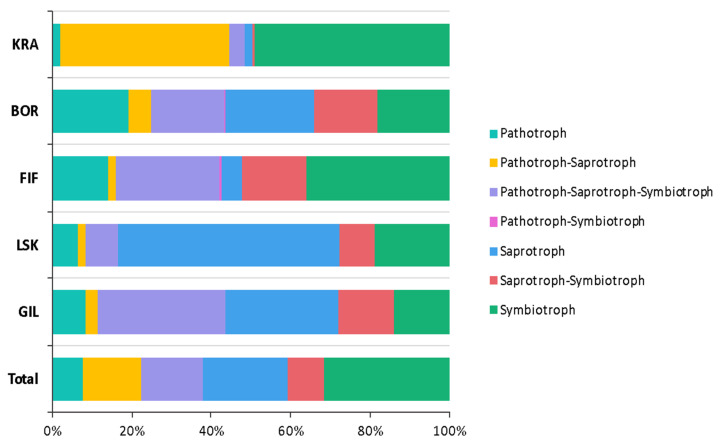
Proportion of different functional groups of biocrust fungi identified by the FUNGuild database.

**Figure 3 microorganisms-09-02195-f003:**
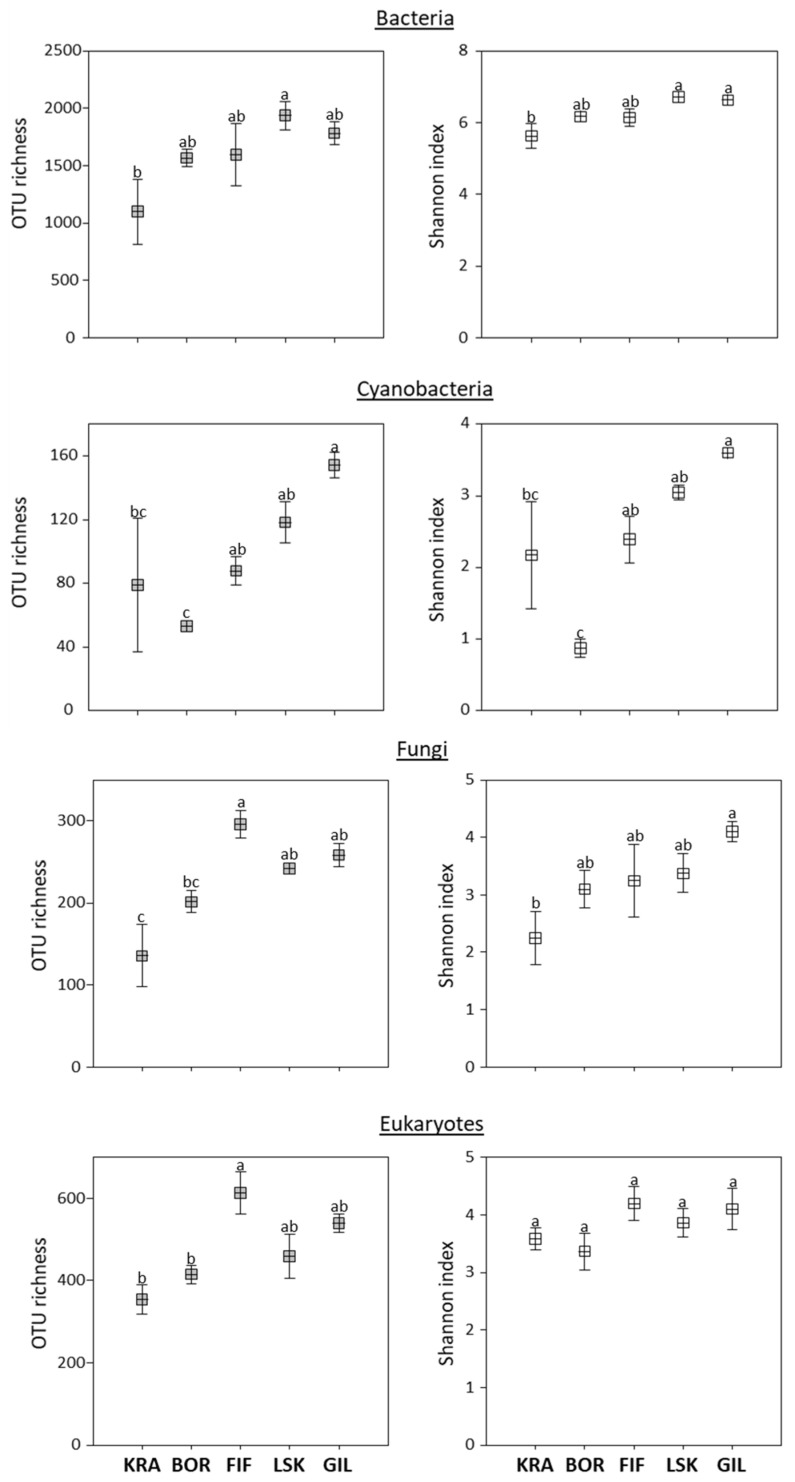
OTU richness and Shannon index in Icelandic biocrusts. The sites are listed in order of elevation gradient from 11 to 157 m a.s.l., respectively. KRA—Krákunes, BOR—Borgarfjarðarbraut, FIF—Fiflholt, LSK—Litla-Skard, and GIL—Giljar. Same letters indicate no statistical difference between the samples according to one-way ANOVA followed by Tukey HSD post hoc test.

**Figure 4 microorganisms-09-02195-f004:**
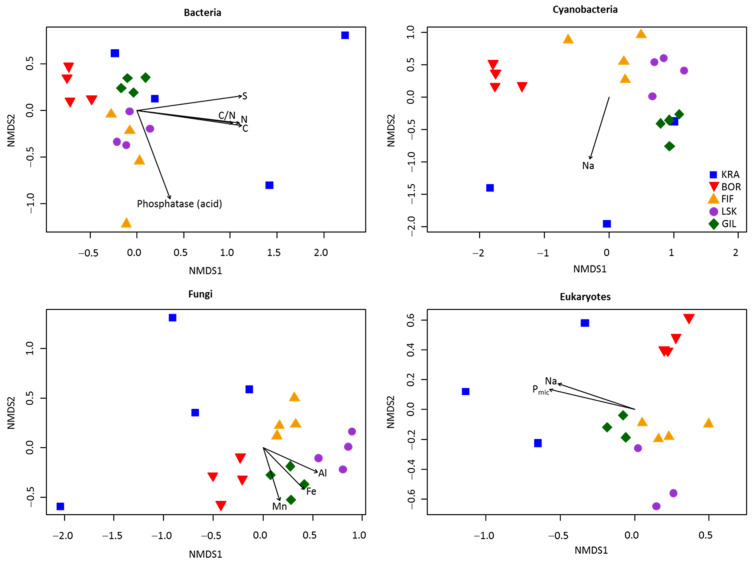
Non-metric multidimensional scaling (NMDS) plots based on relative abundance of the OTUs in the biocrusts. Arrows indicate significant correlations (*p* < 0.05) with environmental variables. Phosphatase (acid), P_mic_, C, and N refer to the potential activity of acid phosphatase, microbial P, total organic carbon, and total organic nitrogen, respectively.

**Table 1 microorganisms-09-02195-t001:** Cyanobacteria (**a**) and eukaryotic microalgae (**b**) isolated from the localities and their most closely related isolates retrieved from the NCBI database.

(**a**)		
**Code**	**Site**	**Nearest Cultivated Neighbor from NCBI Database (ID%)**
KRA-1a	KRA	MK211225.1 *Microcoleus vaginatus* KZ-2-2-5 (94.8%)
KRA-4b		MK211225.1 *Microcoleus vaginatus* KZ-2-2-5 (95.1%)
BOR-4c	BOR	KU219728.1 *Phormidesmis arctica* HOR 11-2 (99.9%)
FIF-1a	FIF	KU219728.1 *Phormidesmis arctica* HOR 11-2 (99.8%)
FIF-1d		MK211225.1 *Microcoleus vaginatus* KZ-2-2-5 (95.3%)
LSK-4c	LSK	MK211225.1 *Microcoleus vaginatus* KZ-2-2-5 (97.9%)
GIL-2a	GIL	KJ939033.1 *Phormidesmis* sp. WJT36-NPBG15 (97.9%)
GIL-2b		MN158649.1 *Wilmottia murrayi* ACKU582 (98.3%)
GIL-2c		MK211225.1 *Microcoleus vaginatus* KZ-2-2-5 (98.1%)
GIL-3b		KU219738.1 *Phormidesmis arctica* MUM 11-8 (97.5%)
(**b**)		
**Code**	**Site**	**Nearest Cultivated Neighbor from NCBI Database (ID%)**
KRA-2b	KRA	JQ315503.1 *Chlamydomonas hedleyi* KMMCC 188 (99.9%)
KRA-4a		MK541759.1 *Gonium pectorale* CCAP 32/23 (99.3%)
BOR-1c	BOR	AB936289.1 *Chlorococcum lobatum* (99.0%)
BOR-3b		AY846370.1 *Ankistrodesmus fusiformis* Itas 8/18 M-7w (99.0%)
BOR-3c		MN267184.1 *Xerochlorella olmae* UTEX B 2993 (99.3%)
BOR-4a		MN248530.1 *Chlorella vulgaris* CCAP 211/21B (100%)
FIF-1b	FIF	MN267184.1 *Xerochlorella olmae* UTEX B2993 (100%)
FIF-1c		MK541792.1 *Chlorella vulgaris* CCAP 211/19 (100%)
FIF-2a		MH703776.1 *Chlamydomonas callunae* SAG 68.81 (99.8%)
FIF-3a		MK262904.1 *Klebsormidium elegans* ACSSI 187 (100%)
FIF-3b		MN248530.1 *Chlorella vulgaris* CCAP 211/21B (99.9%)
FIF-4a		LT560349.1 *Elliptochloris subsphaerica* SAG 2202 (100%)
LSK-1a	LSK	MN267184.1 *Xerochlorella olmae* UTEX B 2993 (100%)
LSK-1b		KM020100.1 *Chlorococcum citriforme* SAG 62.80 (100%)
LSK-1c		KF673372.1 *Chlorella emersonii* SAG 2337 (99.9%)
LSK-4a		MN248530.1 *Chlorella vulgaris* CCAP 211/21B (100%)
LSK-4b		MT425956.1 *Heterochlamydomonas* sp. ACSSI 328 (99.9%)
GIL-3a	GIL	KJ756841.1 *Stichococcus bacillaris* CCAP 379/5 (99.0%)
GIL-3d		MG696565.1 *Nannochloris* sp. ACSSI 144 (100%)
GIL-4a		EU878373.1 *Parietochloris alveolaris* UTEX 836 (99.3%)

**Table 2 microorganisms-09-02195-t002:** Differences in microbial community composition among five sites revealed by PERMANOVA (9999 permutations). Asterisks corresponds to the level of significance (*** *p* < 0.001). Pairwise comparisons are shown in Appendix A.

	r^2^	*F*
Bacteria	0.45	3.07 ***
Cyanobacteria	0.58	4.75 ***
Fungi	0.41	2.59 ***
Eukaryotes	0.54	3.47 ***

## Data Availability

Sequence contigs of isolates were deposited in GenBank with accession numbers MZ020203–MZ020221 for eukaryotic microalgae and MZ020189–MZ020202 for cyanobacteria. The raw reads from amplicon sequencing were submitted to the European Nucleotide Archive (ENA) under the project PRJEB45587.

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
