# Peer review of "Microbial Diversity in Subarctic Biocrusts from West Iceland following an Elevation Gradient"

_microorganisms, 2021, doi:10.3390/microorganisms9112195_

Round 1

Reviewer 1 Report

First major point: This paper should not have been released for peer review because it lacks line numbers. How am I supposed to guide you to the points I am about to make? I have copied text from the paper to here and commented. I’m afraid you will have to find the text on the page… at least I divided my comments for each page… This is a well written description of microbial diversity over an elevation gradient. It is generally well written in the English sense, too, but there are some curious mistakes and conflicts which are easily resolved. Page 1 The title does not describe the work done. “An overview of microbial diversity in subarctic biocrusts from west Iceland based on amplicon metagenomic sequencing.” ‘Overview’ is irrelevant, and we could argue if this work is an overview or a more thorough analysis. You also refer repeatedly to cultures, so why not include that approach in the title? You even have a section in the methods about culturing, “2.2. Cultivation and molecular analyses of microbial phototrophs.” The title would be more descriptive like this (suggestion): “Microbial diversity in west Iceland subarctic biocrusts.” Or, “Microbial diversity in west Iceland subarctic biocrusts over an elevation gradient.” These retain the main points. The keywords can include ‘culturing’ or ‘cultivation’, and ‘high-throughput sequencing’. Your address is something you’ve had for years, probably, but “Research Group Translational Bioinformatics” sounds odd. Are you sure it’s not “Translational Bioinformatics Research group”… “gap of knowledge” should be “gap in Knowledge” ”stretching away from the sea coast” should be “perpendicular to the coast” (perpendicular means at 90 degrees to, or right angles) Four groups of organisms were targeted: bacteria and cyanobacteria (16S rRNA gene), (transcribed spacer region [ITS]) and other eukaryotes (18S rRNA gene). Are the 16S primers for ‘bacteria’ specific to a domain, or do they target Archaea and Bacteria (upper case first letters, italics, domain names)? If both domains, you should make that clear, like this “Four groups of organisms were targeted: bacteria (Archaea and Bacteria), and separately, cyanobacteria (16S rRNA gene), plus fungi (internal transcribed spacer region [ITS]), and other eukaryotes (18S rRNA gene).” Obviously cyanobacteria are Bacteria, so you can say ‘separately’ here to show you weren’t using the same primers for 16S that target the (other) bacteria (Archaea and/or Bacteria). “filamentous orders Synechococcales”… Synechococcales includes non-filamentous forms. You should not say this order is filamentous. It implies all are filamentous. “phototrophs”. Don’t used this word. The term is photoautotroph. This should be replaced throughout the manuscript. Phototroph means ‘to eat light’! Phormidesmis, Microcoleus, Wilmottia and Oscillatoria… you didn’t italicize any genus or species names, except in a table. Please correct this. In general, the microbial diversity in the… What do you mean by microbial diversity? You should explain, such as ‘taxonomic’, metabolic, physiological. Not everyone even agrees what is ‘microbial diversity’, so let’s make it clear from the start. You also don’t need a definite article (the) before ‘microbial diversity’. increased following the altitude gradient = along the elevation gradient strong temperature and light amplitudes. Amplitudes doesn’t sound like the right word. Does temperature even have an amplitude? Do you mean gradients? Such as very cold to very hot over time. And irradiation can also vary substantially over time, and obviously with depth in the biocrust. So, you probably need “…strong temperature and light gradients”. CO2 – subscript the 2 (everywhere you have this) biocrust development forming = biocrust development by forming

Page 2 Non-phototrophic = Non-photoautotrophic competitive advantages and multiple… insert a comma after advantages. UV repair mechanisms… insert a comma after mechanisms Phototrophs into lichens – photoautotrophs tolerance to freeze-thaw cycles = tolerance of freeze-thaw cycles are very distinct = are distinct. No need for ‘very’ slow down = slow. No need for ‘down’ plant colonization in Iceland and – insert a comma after Iceland “Furthermore, arbuscular mycorrhizal fungi [15,16] and other terrestrial fungi [17,18] were described in Icelandic soils using morphological methods. Similarly, diversity of cyanobacteria and eukaryotic microalgae in Icelandic soils was only reported in papers from the last century [19,20].” This paragraph is a little irritating to read, with four consecutive sentences starting with, for example, likewise, furthermore, and similarly. It’s too much! How about this: “Furthermore, arbuscular mycorrhizal fungi [15,16] and other terrestrial fungi [17,18] were described in Icelandic soils using morphological methods, while the diversity of cyanobacteria and eukaryotic microalgae was only reported more recently [19,20].” Italicize all genus names. our recent study [12] based… put the [12] at the end of the sentence. Your site description is weak. You mention a temperature range, but do you have climate data for each site along your transect? I mention that because I just don’t imagine there’s much difference in temperature over what seems to be a small elevation change (156 – 11 = 145 m). Specifically, is there a significant difference in temperature between 11 m and 156 m? On the other hand, I can envisage a temperature change, or difference, with increasing distance from the sea, such as ‘warm’ at the coast but colder inland. It seems distance from the sea is more important for temperature than elevation, that’s all… further north from Reykjavik and belongs to the National = “…north of Reykjavik, belonging to the National…

Page 3 The sampling = Sampling It was brought = delivered sea into inland = sea to inland where it was stored in a freezer – where they were stored in a freezer are present in Pushkareva et al. [12] = are reported elsewhere [12]. high content of P = high P content where moderately labile P was dominant. = and moderately labile P was dominant. Microphototrophic = Microphotoautotrophic Italicize all genus and species names In addition, majority of fungal strains = In addition, most of the fungal strains from the studied biocrusts = from the biocrusts [‘studied’ is superfluous] photons m−2 s−1 = superscript the -2 and the -1. DNA of microphototrophic strains = DNA of microphotoautotrophic strains. What does this mean? Micro in this case might mean ‘low levels of light’. Think of micro in microaerophile. Are you implying your strains require ‘micro’ levels of light? Or are you saying they are microbial? I think for the latter you should call them ‘microbial photoautotrophs’, because that is clear. It is not a term about how much light they prefer. DNA of microphototrophic strains was extracted using NucleoSpin Plant II mini kit = DNA was extracted from microbial photoautotroph in the NucleoSpin Plant II mini kit… Superscript all degree symbols sequencing using Illumina MiSeq = sequencing using the Illumina MiSeq “Sequences of isolates were generated using the software Geneious 8.1.9…” Are you sure? Sequences are output by the Illumina MiSeq, in a form. These data are processed through Geneious, in your case. “The most closely related isolates were fetched from GenBank using BLAST.” Not really. Isolates are cultures. You didn’t retrieve those from BLAST. What you did was compare your sequences with other sequences in GenBank through BLAST searches (Altschul et al., 1997). Now, if you configure the search you can have BLAST only compare your sequences with type material and exclude environmental samples. Type strains will give you the best matches because they are the best curated. If you BLAST against everything you can end up with all kinds of nonsensical matches that have been mislabeled over the years. Unless the GenBank staff has culled all those badly labeled sequences now. You should rewrite the sentence and describe what you compared your sequences with.

Page 4 A phylogenetic trees = Phylogenetic trees/A phylogenetic tree (plural/singular) their best GenBank hits – what is a best hit? Do you mean the closest neighboring sequences, or closest matching sequences? removed for further analyses = removed from further analyses not included to alpha and beta = not included in alpha and beta groups according their nutrient acquisition = groups according to their nutrient acquisition reads in total for 20 samples = reads for 20 samples

Page 5 pathway in 16S dataset = pathway in the 16S dataset

Page 6 Italicize genus names A total of 364233 reads and 624 OTUs assigned to Fungi were obtained by sequencing of ITS region = “A total of 364233 reads and 624 OTUs assigned to Fungi were obtained by sequencing part of the ITS region… from order Synechococcales (> 93% similarity). = “from the order Synechococcales (> 93% similarity).” And by the way, this is not really similarity; it is ‘identity’. We should be talking about sequence identity because it’s nucleotide for nucleotide. We use similarity to describe amino acid sequences of proteins, when different amino acids provide the same function, or behave in the same way (similar). In a nucleic acid, the nucleotide is either identical or it is not. So, everywhere you talk about sequence similarity is actually nucleotide identity. Just because most other people say ‘similarity’ doesn’t mean it’s right… “Class Agaricomycetes was the most dominant within Basidiomycota with 11% of total fungal reads.” = Sequences that affiliated with the Class Agaricomycetes comprised the majority of those from the Basidiomycota, with 11% of total ITS reads. (ITS, right? This is based on your ITS, not ‘other eukaryotes’ or 18S?) …sequencing of 18S rRNA constituted = …sequencing of part of the 18S rRNA gene constituted and 58% of Opisthokonta clade. = and 58% of those Opisthokonta clade. 3.1.4 Sequencing of 18S rRNA gene produced… You didn’t sequenced the 18S rRNA gene. You sequenced part of it. Indeed, didn’t you sequence amplified fragments of the 18S rRNA genes. The majority of reads belonged to clade… = “The majority of the reads affiliated with…”

Page 7 Archaeplastida was presented by the clade Chloroplastida only, which = The Archaeplastida was represented by the clade Chloroplastida only, which two algal phyla Charophyta (80 OTUs) and Chlorophyta (245 OTUs). = two algal phyla [comma] Charophyta (80 OTUs) [comma] and Chlorophyta (245 OTUs). 30% (105 OTUs) of Stramenopiles, 34 % (109 OTUs) of Al-veolata and 36% (211 OTUs) of Rhizaria. = 30% of 105 OTUs affiliated with the Stramenopiles, 34 % (109) affiliated with the Alveolata, and 36% (211) affiliated with the Rhizaria.“ Additionally, 20 strains of eukaryotic microalgae were isolated and sequenced… Really, genome sequenced? 18S sequenced? Fragment of the 18S? What did you sequence? Italicize genus names Table 1. Cyanobacteria (a) and eukaryotic microalgae (b) isolated from the localities and their most closely related isolates fetched from NCBI database. They were not ‘fetched’. They were retrieved, as described earlier. The closest NCBI isolate sequence (ID%) = Nearest cultivated neighbor based on sequence identity (%)

Page 8 increase following altitudinal gradient from the sea to the inland with some = increase over the elevation gradient from the sea, with some… Besides, site BOR exhibited an apparent distinct community structure compared to the other sites (Fig. 1). For example, there was significantly higher percentage of reads be-longing to Proteobacteria, cyanobacterial order Synechococcales and clade Archaeplas-tida than in biocrusts from the other sites. Likewise, the majority of reads at site BOR were not assigned to any fungal phyla, which was also discrete from other sites. Besides, relative abundance of pathotrophs was higher there than at the other sites (Fig. 2). I think this is what you are trying to say: “Further, community structure at the BOR site appeared distinct from those at the other sites (Fig. 1). For example, significantly more reads from BOR affiliated with Proteobacteria, the Synechococcales, and the Archaeplastida than did those at other sites. Similarly, most reads from the BOR site did not affiliate with a fungal phylum, and the highest relative abundance of pathotrophs was found here (Fig. 2).” Table 2. Differences of microbial community composition between the five studied sites based on PERMANOVA (9999 permutations). Asterisks corresponds to the level of significance = Table 2. Differences in microbial community composition between five sites, based on PERMANOVA (9999 permutations). Asterisks correspond to the level of significance

Page 9 OTU richness and Shannon index in the studied Icelandic biocrusts. The sites are listed in order of altitudinal gradient = OTU richness and Shannon index in Icelandic biocrusts. Sites are listed in their order over the elevational gradient, with lowest first. and potential activity of acid phosphatase. How did you get AP activity? Is that in the methods? I searched the file for phosphatase and this was the first mention. The same applies to “Na and microbial P”. If these are from another paper, you should cite the paper.

Page 10: soil chemical parameters = soil chemistry parameters

Page 11: 4. Discussions = 4. Discussion right below the Arctic Circle, received far less attention = right below the Arctic Circle, has received far less attention Overall, the bacterial community in the studied Icelandic biocrusts was mainly = Overall, bacterial communities in the biocrusts described here comprised largely… Actinobacteria (Fig. 1a) which is consistent = Actinobacteria, which is consistent with… [and put (Fig. 1a) at the end of the sentence, before the citation.] maintenance due to the production of exopolysaccharides = maintenance due to their production of exopolysaccharides (Proteobacteria) or mycelium (Actinobacteria) = (Proteobacteria) or mycelia (Actinobacteria) high organic matter promote Proteobacteria = high organic matter content promote Proteobacteria. (And promote what, in Proteobacteria? Respiratory activity? Biofilm formation? Growth? Biofilm formation? We can just say something promotes humans, for example. Or promotes yeast, etc. What is being promoted? “…which might explain their dominance in the studied Icelandic biocrusts with Andosols soil type.” That’s a little too simple, and offers no insight. It’s like, “Something promotes them, and that’s why they were dominant!” Well, of course. What other mechanism(s) might explain their ‘dominance’ (in terms of numbers of sequences, by the way, no other measure)? Maybe the wind brought them here in dust? All you have are sequences and sequence numbers, after all. However, if they’re often observed in permafrost I would also expect them to be tolerant of cold conditions. Unless they’ve been there 10,000 years since it was last warm in the area. In short, finally, I’d delete this part of the sentence, “…which might explain their dominance in the studied Icelandic biocrusts with Andosols soil type.” “…soils, strongly participate in the nitrogen cycle by performing an anaerobic ammonium oxidation…” But you don’t know that for here. That is just speculation. Did you measure anaerobic ammonium oxidation, for example? Do you even know if it’s an ongoing process in your sites? The studied biocrusts were dominated… You don’t have to keep saying ‘the studied’. If you didn’t study them you would not have any data. You are presenting data, so we know they were ‘studied’. Just like you don’t always have to label a samples that was collected from soil as ‘the sample soil’, or the soil sample’. It’s obvious. Italicize Leptolyngbya (>97% similarity). 1. It's identity. 2) Why do you use 97%? Typical nowadays in taxonomy is 98.6%; two sequences sharing >98.6% nucleotide identity in the 16S rRNA gene are said to come from the same species. Try a paper or two by Stackebrandt. And others now, such as for genome to genome distances, and how they relate to 16S sequence identities. “This might indicate that the database used for the identification of the reads obtained by amplicon sequencing is outdated and could provide incorrect taxonomic assignments on the genus level.” It tells you nothing of the sort! The database is as good as the sequences uploaded, and how accurately they are named. Maybe the sequence you matched with is either from another organism but has been wrongly labeled as a Phormidesmis, or would you believe there are cyanobacteria outside Leptolyngbya that are filamentous? Just because it’s filamentous you assume it’s a Leptolyngbya? I’ve BLASTed a sequence and the first match was a Bacillus, but every other match was a Vibrio. Someone had wrongly named their ‘Bacillus’ sequence, meaning it was really a Vibrio but for some reason they called it ‘Bacillus’ when entering the sequence details. So, either you found a real Phormidesmis, or something close. You maybe didn’t find a Leptolyngbya, though. Even if you think it looks like one. How about the Fischerella? They’re filamentous, too, but not Leptolyngbya. These fungi acquire ecophysiological traits for survival = These fungi have evolved ecophysiological traits that permit survival… Unless you can show they acquired the genes involved from someone else. “Besides, the high number of OTUs, which could not be assigned to any phyla, might indicate either database limitation or yet undescribed fungal species in Icelandic biocrusts.” This is not database limitation. Well, it is, because every organism on Earth is not represented. However, why not just try to emphasize the likelihood that the communities comprise novel microbes? That’s acceptable, and also likely.

Page 12 Archaeplastida presented only by Chlorophyta and Charophyta prevailed other eu-karyotes in the studied biocrusts = Archaeplastida, represented here only by Chlorophyta and Charophyta, numerically dominated the eukaryote communities in terms of the number of sequences representing each biocrust. “…confirming that a large part of microalgae remains uncultivable.” It does not confirm that at all! That is an outrageous assertion. It merely confirms you did not use the right medium or other cultivation conditions. If you have infinite resources and infinite time you will cultivate a representative of every microbe on Earth. You have none of those, so you cannot cultivate them all. However, that will not confirm they are ‘uncultivable’. It only confirms you did not use the right medium or other cultivation conditions, and that you do not have infinite resources and infinite time. Right? “There were not many OTUs belonging to yellow-green algae Xanthophyceae detected by amplicon sequencing, which reflected in the absence of isolates of this class.” Oh. Is that interesting or relevant? It does not strike me as interesting or relevant: I could expand on that, but I’m already typing on the sixth page, so let’s just say you should delete it. were similarly detected in low abundance = were also detected in low abundance Furthermore, phylum Cercozoa (Rhizaria), a typical soil protist group, was the dominant representative of SAR clade in the studied biocrusts. These single-celled eukaryotes have been already de-scribed in the Icelandic biocrusts, showing the dominance of order Glissomonadida [50], = Furthermore, the phylum Cercozoa (Rhizaria), a typical soil protist group, was the dominant representative of the SAR clade in the biocrusts. These single-celled eukaryotes have been already described here, with members of the order Glissomonadida dominating [50]… This finding supports the study, where the negative effect of salt in the coastal areas on soil microbial community composition has been described [52]. = This supports a previous finding that salt in the coastal area negatively impacted soil microbial community composition [52]. KRA is also in agreement with = KRA is consistent with site KRA were more than 99% similar to = site KRA shared >99% sequence identity with… located further away from the sea but in low altitude, was = located further from the sea but at lower elevation, was… filamentous cyanobacteria Synechococcales = filamentous Synechococcales other studied sites, where phylum Chlorophyta = other studied sites, where members of the phylum Chlorophyta…

Page 13 In addition, glutathione metabolism = In addition, a glutathione metabolism several bacterial genera and majority of them were = several bacterial genera, and most of them were… glutathione metabolism at the site BOR with lowest carbon and nitrogen contents as well as C/N ratio = glutathione metabolism at BOR, with the lowest carbon and nitrogen contents, and C/N ratio. been provided before for the subarctic island Iceland. Furthermore, we observed the overall increase of microbial diversity with altitude rise. = …been provided for Iceland. Furthermore, we observed an increase in microbial diversity with increasing elevation.

Author Response

Reviewer 1.

First major point: This paper should not have been released for peer review because it lacks line numbers. How am I supposed to guide you to the points I am about to make? I have copied text from the paper to here and commented. I’m afraid you will have to find the text on the page… at least I divided my comments for each page…

Re: We are apologizing for this inconvenience. The function “Line numbers” was used but because of the special formatting of the journal template, the line numbers were not displayed.

This is a well written description of microbial diversity over an elevation gradient. It is generally well written in the English sense, too, but there are some curious mistakes and conflicts which are easily resolved.

Re: We would like to thank you for providing such a detailed review that helped a lot to improve this manuscript.

Page 1 The title does not describe the work done. “An overview of microbial diversity in subarctic biocrusts from west Iceland based on amplicon metagenomic sequencing.” ‘Overview’ is irrelevant, and we could argue if this work is an overview or a more thorough analysis. You also refer repeatedly to cultures, so why not include that approach in the title? You even have a section in the methods about culturing, “2.2. Cultivation and molecular analyses of microbial phototrophs.” The title would be more descriptive like this (suggestion): “Microbial diversity in west Iceland subarctic biocrusts.” Or, “Microbial diversity in west Iceland subarctic biocrusts over an elevation gradient.” These retain the main points.

Re: Thank you for the title suggestions. We modified it into “Microbial diversity in subarctic biocrusts from west Iceland following an elevation gradient”.

The keywords can include ‘culturing’ or ‘cultivation’, and ‘high-throughput sequencing’.

Re: The suggested keywords were added.

Your address is something you’ve had for years, probably, but “Research Group Translational Bioinformatics” sounds odd. Are you sure it’s not “Translational Bioinformatics Research group”…

Re: The name of the group is correct.

“gap of knowledge” should be “gap in Knowledge” ”stretching away from the sea coast” should be “perpendicular to the coast” (perpendicular means at 90 degrees to, or right angles)

Re: Modified as suggested.

Four groups of organisms were targeted: bacteria and cyanobacteria (16S rRNA gene), (transcribed spacer region [ITS]) and other eukaryotes (18S rRNA gene). Are the 16S primers for ‘bacteria’ specific to a domain, or do they target Archaea and Bacteria (upper case first letters, italics, domain names)? If both domains, you should make that clear, like this “Four groups of organisms were targeted: bacteria (Archaea and Bacteria), and separately, cyanobacteria (16S rRNA gene), plus fungi (internal transcribed spacer region [ITS]), and other eukaryotes (18S rRNA gene).” Obviously cyanobacteria are Bacteria, so you can say ‘separately’ here to show you weren’t using the same primers for 16S that target the (other) bacteria (Archaea and/or Bacteria).

Re: Archaea were not included in this study because the primers used targeted bacteria. Later in the methods we separate cyanobacteria and bacteria and explain why.

“filamentous orders Synechococcales”… Synechococcales includes non-filamentous forms. You should not say this order is filamentous. It implies all are filamentous.

Re: The collocation was modified into “filamentous forms from orders Synechococcales and Oscillatoriales”.

“phototrophs”. Don’t used this word. The term is photoautotroph. This should be replaced throughout the manuscript. Phototroph means ‘to eat light’!

Re: Replaced as suggested

Phormidesmis, Microcoleus, Wilmottia and Oscillatoria… you didn’t italicize any genus or species names, except in a table. Please correct this.

In general, the microbial diversity in the… What do you mean by microbial diversity? You should explain, such as ‘taxonomic’, metabolic, physiological. Not everyone even agrees what is ‘microbial diversity’, so let’s make it clear from the start. You also don’t need a definite article (the) before ‘microbial diversity’. increased following the altitude gradient = along the elevation gradient strong temperature and light amplitudes.  Amplitudes doesn’t sound like the right word. Does temperature even have an amplitude? Do you mean gradients? Such as very cold to very hot over time. And irradiation can also vary substantially over time, and obviously with depth in the biocrust. So, you probably need “…strong temperature and light gradients”. CO2 – subscript the 2 (everywhere you have this) biocrust development forming = biocrust development by forming

Re: Corrected as suggested.

Page 2 Non-phototrophic = Non-photoautotrophic competitive advantages and multiple… insert a comma after advantages. UV repair mechanisms… insert a comma after mechanisms Phototrophs into lichens – photoautotrophs tolerance to freeze-thaw cycles = tolerance of freeze-thaw cycles are very distinct = are distinct. No need for ‘very’ slow down = slow. No need for ‘down’ plant colonization in Iceland and – insert a comma after Iceland “Furthermore, arbuscular mycorrhizal fungi [15,16] and other terrestrial fungi [17,18] were described in Icelandic soils using morphological methods. Similarly, diversity of cyanobacteria and eukaryotic microalgae in Icelandic soils was only reported in papers from the last century [19,20].” This paragraph is a little irritating to read, with four consecutive sentences starting with, for example, likewise, furthermore, and similarly. It’s too much! How about this: “Furthermore, arbuscular mycorrhizal fungi [15,16] and other terrestrial fungi [17,18] were described in Icelandic soils using morphological methods, while the diversity of cyanobacteria and eukaryotic microalgae was only reported more recently [19,20].” Italicize all genus names. our recent study [12] based… put the [12] at the end of the sentence.

Re: Corrected as suggested.

Your site description is weak. You mention a temperature range, but do you have climate data for each site along your transect? I mention that because I just don’t imagine there’s much difference in temperature over what seems to be a small elevation change (156 – 11 = 145 m). Specifically, is there a significant difference in temperature between 11 m and 156 m? On the other hand, I can envisage a temperature change, or difference, with increasing distance from the sea, such as ‘warm’ at the coast but colder inland. It seems distance from the sea is more important for temperature than elevation, that’s all… further north from Reykjavik and belongs to the National = “…north of Reykjavik, belonging to the National…

Re: We added more information to the “site description” section. Unfortunately, we do not have information about possible microclimatic temperature gradients as there are no meteorological stations on the sampling sites. But we added some additional factors (precipitation) from the next climate station Borgarnes.

Page 3 The sampling = Sampling It was brought = delivered sea into inland = sea to inland where it was stored in a freezer – where they were stored in a freezer are present in Pushkareva et al. [12] = are reported elsewhere [12]. high content of P = high P content where moderately labile P was dominant. = and moderately labile P was dominant. Microphototrophic = Microphotoautotrophic Italicize all genus and species names In addition, majority of fungal strains = In addition, most of the fungal strains from the studied biocrusts = from the biocrusts [‘studied’ is superfluous] photons m−2 s−1 = superscript the -2 and the -1. DNA of microphototrophic strains = DNA of microphotoautotrophic strains. What does this mean? Micro in this case might mean ‘low levels of light’. Think of micro in microaerophile. Are you implying your strains require ‘micro’ levels of light? Or are you saying they are microbial? I think for the latter you should call them ‘microbial photoautotrophs’, because that is clear. It is not a term about how much light they prefer. DNA of microphototrophic strains was extracted using NucleoSpin Plant II mini kit = DNA was extracted from microbial photoautotroph in the NucleoSpin Plant II mini kit… Superscript all degree symbols sequencing using Illumina MiSeq = sequencing using the Illumina MiSeq “Sequences of isolates were generated using the software Geneious 8.1.9…” Are you sure? Sequences are output by the Illumina MiSeq, in a form. These data are processed through Geneious, in your case. “The most closely related isolates were fetched from GenBank using BLAST.” Not really. Isolates are cultures. You didn’t retrieve those from BLAST. What you did was compare your sequences with other sequences in GenBank through BLAST searches (Altschul et al., 1997). Now, if you configure the search you can have BLAST only compare your sequences with type material and exclude environmental samples. Type strains will give you the best matches because they are the best curated. If you BLAST against everything you can end up with all kinds of nonsensical matches that have been mislabeled over the years. Unless the GenBank staff has culled all those badly labeled sequences now. You should rewrite the sentence and describe what you compared your sequences with.

Re: Corrected as suggested.

Page 4 A phylogenetic trees = Phylogenetic trees/A phylogenetic tree (plural/singular) their best GenBank hits – what is a best hit? Do you mean the closest neighboring sequences, or closest matching sequences? removed for further analyses = removed from further analyses not included to alpha and beta = not included in alpha and beta groups according their nutrient acquisition = groups according to their nutrient acquisition reads in total for 20 samples = reads for 20 samples

Re: Corrected as suggested.

Page 5 pathway in 16S dataset = pathway in the 16S dataset

Re: Corrected as suggested.

Page 6 Italicize genus names A total of 364233 reads and 624 OTUs assigned to Fungi were obtained by sequencing of ITS region = “A total of 364233 reads and 624 OTUs assigned to Fungi were obtained by sequencing part of the ITS region… from order Synechococcales (> 93% similarity). = “from the order Synechococcales (> 93% similarity).” And by the way, this is not really similarity; it is ‘identity’. We should be talking about sequence identity because it’s nucleotide for nucleotide. We use similarity to describe amino acid sequences of proteins, when different amino acids provide the same function, or behave in the same way (similar). In a nucleic acid, the nucleotide is either identical or it is not. So, everywhere you talk about sequence similarity is actually nucleotide identity. Just because most other people say ‘similarity’ doesn’t mean it’s right… “Class Agaricomycetes was the most dominant within Basidiomycota with 11% of total fungal reads.” = Sequences that affiliated with the Class Agaricomycetes comprised the majority of those from the Basidiomycota, with 11% of total ITS reads. (ITS, right? This is based on your ITS, not ‘other eukaryotes’ or 18S?) …sequencing of 18S rRNA constituted = …sequencing of part of the 18S rRNA gene constituted and 58% of Opisthokonta clade. = and 58% of those Opisthokonta clade.

Re: Corrected as suggested.

3.1.4 Sequencing of 18S rRNA gene produced… You didn’t sequenced the 18S rRNA gene. You sequenced part of it. Indeed, didn’t you sequence amplified fragments of the 18S rRNA genes.

Re: Yes, it is, indeed, true. However, as we already gave an explanation that we sequenced only part of the 18S, we would like to leave it here in this state because of easier understanding. If the reviewer disagrees, we will change it accordingly.

The majority of reads belonged to clade… = “The majority of the reads affiliated with…”

Page 7 Archaeplastida was presented by the clade Chloroplastida only, which = The Archaeplastida was represented by the clade Chloroplastida only, which two algal phyla Charophyta (80 OTUs) and Chlorophyta (245 OTUs). = two algal phyla [comma] Charophyta (80 OTUs) [comma] and Chlorophyta (245 OTUs). 30% (105 OTUs) of Stramenopiles, 34 % (109 OTUs) of Al-veolata and 36% (211 OTUs) of Rhizaria. = 30% of 105 OTUs affiliated with the Stramenopiles, 34 % (109) affiliated with the Alveolata, and 36% (211) affiliated with the Rhizaria.“ Additionally, 20 strains of eukaryotic microalgae were isolated and sequenced… Really, genome sequenced? 18S sequenced? Fragment of the 18S? What did you sequence? Italicize genus names Table 1. Cyanobacteria (a) and eukaryotic microalgae (b) isolated from the localities and their most closely related isolates fetched from NCBI database. They were not ‘fetched’. They were retrieved, as described earlier. The closest NCBI isolate sequence (ID%) = Nearest cultivated neighbor based on sequence identity (%)

Re: Corrected as suggested.

Page 8 increase following altitudinal gradient from the sea to the inland with some = increase over the elevation gradient from the sea, with some… Besides, site BOR exhibited an apparent distinct community structure compared to the other sites (Fig. 1). For example, there was significantly higher percentage of reads be-longing to Proteobacteria, cyanobacterial order Synechococcales and clade Archaeplas-tida than in biocrusts from the other sites. Likewise, the majority of reads at site BOR were not assigned to any fungal phyla, which was also discrete from other sites. Besides, relative abundance of pathotrophs was higher there than at the other sites (Fig. 2). I think this is what you are trying to say: “Further, community structure at the BOR site appeared distinct from those at the other sites (Fig. 1). For example, significantly more reads from BOR affiliated with Proteobacteria, the Synechococcales, and the Archaeplastida than did those at other sites. Similarly, most reads from the BOR site did not affiliate with a fungal phylum, and the highest relative abundance of pathotrophs was found here (Fig. 2).” Table 2. Differences of microbial community composition between the five studied sites based on PERMANOVA (9999 permutations). Asterisks corresponds to the level of significance = Table 2. Differences in microbial community composition between five sites, based on PERMANOVA (9999 permutations). Asterisks correspond to the level of significance

Page 9 OTU richness and Shannon index in the studied Icelandic biocrusts. The sites are listed in order of altitudinal gradient = OTU richness and Shannon index in Icelandic biocrusts. Sites are listed in their order over the elevational gradient, with lowest first. and potential activity of acid phosphatase.

Re: Corrected as suggested.

How did you get AP activity? Is that in the methods? I searched the file for phosphatase and this was the first mention. The same applies to “Na and microbial P”. If these are from another paper, you should cite the paper.

Re: These parameters were measured and reported in our previous publication (Pushkareva et al. 2021). The citation was added to the text “Moreover, community dissimilarities for each microbial group were explained by different environmental parameters, measured in our previous publication [1].”

Page 10: soil chemical parameters = soil chemistry parameters

Page 11: 4. Discussions = 4. Discussion right below the Arctic Circle, received far less attention = right below the Arctic Circle, has received far less attention Overall, the bacterial community in the studied Icelandic biocrusts was mainly = Overall, bacterial communities in the biocrusts described here comprised largely… Actinobacteria (Fig. 1a) which is consistent = Actinobacteria, which is consistent with… [and put (Fig. 1a) at the end of the sentence, before the citation.] maintenance due to the production of exopolysaccharides = maintenance due to their production of exopolysaccharides (Proteobacteria) or mycelium (Actinobacteria) = (Proteobacteria) or mycelia (Actinobacteria)

Re: Corrected as suggested.

high organic matter promote Proteobacteria = high organic matter content promote Proteobacteria. (And promote what, in Proteobacteria? Respiratory activity? Biofilm formation? Growth? Biofilm formation? We can just say something promotes humans, for example. Or promotes yeast, etc. What is being promoted? “…which might explain their dominance in the studied Icelandic biocrusts with Andosols soil type.” That’s a little too simple, and offers no insight. It’s like, “Something promotes them, and that’s why they were dominant!” Well, of course. What other mechanism(s) might explain their ‘dominance’ (in terms of numbers of sequences, by the way, no other measure)? Maybe the wind brought them here in dust? All you have are sequences and sequence numbers, after all. However, if they’re often observed in permafrost I would also expect them to be tolerant of cold conditions. Unless they’ve been there 10,000 years since it was last warm in the area. In short, finally, I’d delete this part of the sentence, “…which might explain their dominance in the studied Icelandic biocrusts with Andosols soil type.”

Re: The sentence was modified as suggested and second part was deleted. “Moreover, soils with high organic matter content promote growth of Proteobacteria”

 “…soils, strongly participate in the nitrogen cycle by performing an anaerobic ammonium oxidation…” But you don’t know that for here. That is just speculation. Did you measure anaerobic ammonium oxidation, for example? Do you even know if it’s an ongoing process in your sites?

Re: The sentence was expanded to point out that it is just a speculation.

The studied biocrusts were dominated… You don’t have to keep saying ‘the studied’. If you didn’t study them you would not have any data. You are presenting data, so we know they were ‘studied’. Just like you don’t always have to label a samples that was collected from soil as ‘the sample soil’, or the soil sample’. It’s obvious. Italicize Leptolyngbya (>97% similarity). 1. It's identity.

Re: Corrected as suggested.

2) Why do you use 97%? Typical nowadays in taxonomy is 98.6%; two sequences sharing >98.6% nucleotide identity in the 16S rRNA gene are said to come from the same species. Try a paper or two by Stackebrandt. And others now, such as for genome to genome distances, and how they relate to 16S sequence identities.

Re: The identity of 97% here corresponds to the closest neighbour retrieved from the NCBI database (based on the sequences of the cultures) and, therefore, just provides a rather rough information.

“This might indicate that the database used for the identification of the reads obtained by amplicon sequencing is outdated and could provide incorrect taxonomic assignments on the genus level.” It tells you nothing of the sort! The database is as good as the sequences uploaded, and how accurately they are named. Maybe the sequence you matched with is either from another organism but has been wrongly labeled as a Phormidesmis, or would you believe there are cyanobacteria outside Leptolyngbya that are filamentous? Just because it’s filamentous you assume it’s a Leptolyngbya? I’ve BLASTed a sequence and the first match was a Bacillus, but every other match was a Vibrio. Someone had wrongly named their ‘Bacillus’ sequence, meaning it was really a Vibrio but for some reason they called it ‘Bacillus’ when entering the sequence details. So, either you found a real Phormidesmis, or something close. You maybe didn’t find a Leptolyngbya, though. Even if you think it looks like one. How about the Fischerella? They’re filamentous, too, but not Leptolyngbya.

Re: We agree with this comment and modified the sentence according to the reviewer explanation. “This might indicate that some sequences stored in the database used for the identification of the reads obtained by amplicon sequencing have incorrect taxonomic names on the genus level.”

These fungi acquire ecophysiological traits for survival = These fungi have evolved ecophysiological traits that permit survival… Unless you can show they acquired the genes involved from someone else. “Besides, the high number of OTUs, which could not be assigned to any phyla, might indicate either database limitation or yet undescribed fungal species in Icelandic biocrusts.” This is not database limitation. Well, it is, because every organism on Earth is not represented. However, why not just try to emphasize the likelihood that the communities comprise novel microbes? That’s acceptable, and also likely.

Re: The sentences were modified as suggested. “Database limitation” was deleted from the text.

Page 12 Archaeplastida presented only by Chlorophyta and Charophyta prevailed other eu-karyotes in the studied biocrusts = Archaeplastida, represented here only by Chlorophyta and Charophyta, numerically dominated the eukaryote communities in terms of the number of sequences representing each biocrust.

Re: Corrected as suggested.

“…confirming that a large part of microalgae remains uncultivable.” It does not confirm that at all! That is an outrageous assertion. It merely confirms you did not use the right medium or other cultivation conditions. If you have infinite resources and infinite time you will cultivate a representative of every microbe on Earth. You have none of those, so you cannot cultivate them all. However, that will not confirm they are ‘uncultivable’. It only confirms you did not use the right medium or other cultivation conditions, and that you do not have infinite resources and infinite time. Right?

Re: We agree with this comment and modified the sentence accordingly. “but were underrepresented in number, pointing that a large part of microalgae remained uncultivable under the conditions we used in this study.”

“There were not many OTUs belonging to yellow-green algae Xanthophyceae detected by amplicon sequencing, which reflected in the absence of isolates of this class.” Oh. Is that interesting or relevant? It does not strike me as interesting or relevant: I could expand on that, but I’m already typing on the sixth page, so let’s just say you should delete it. were similarly detected in low abundance = were also detected in low abundance Furthermore, phylum Cercozoa (Rhizaria), a typical soil protist group, was the dominant representative of SAR clade in the studied biocrusts. These single-celled eukaryotes have been already de-scribed in the Icelandic biocrusts, showing the dominance of order Glissomonadida [50], = Furthermore, the phylum Cercozoa (Rhizaria), a typical soil protist group, was the dominant representative of the SAR clade in the biocrusts. These single-celled eukaryotes have been already described here, with members of the order Glissomonadida dominating [50]… This finding supports the study, where the negative effect of salt in the coastal areas on soil microbial community composition has been described [52]. = This supports a previous finding that salt in the coastal area negatively impacted soil microbial community composition [52]. KRA is also in agreement with = KRA is consistent with site KRA were more than 99% similar to = site KRA shared >99% sequence identity with… located further away from the sea but in low altitude, was = located further from the sea but at lower elevation, was… filamentous cyanobacteria Synechococcales = filamentous Synechococcales other studied sites, where phylum Chlorophyta = other studied sites, where members of the phylum Chlorophyta…

Page 13 In addition, glutathione metabolism = In addition, a glutathione metabolism several bacterial genera and majority of them were = several bacterial genera, and most of them were… glutathione metabolism at the site BOR with lowest carbon and nitrogen contents as well as C/N ratio = glutathione metabolism at BOR, with the lowest carbon and nitrogen contents, and C/N ratio. been provided before for the subarctic island Iceland. Furthermore, we observed the overall increase of microbial diversity with altitude rise. = …been provided for Iceland. Furthermore, we observed an increase in microbial diversity with increasing elevation.

Re: Corrected as suggested.

Reviewer 2 Report

The manuscript describes the microbial diversity in subarctic biological soil crusts at the western Iceland revealed by molecular approaches. The topic is ecologically interesting and important, but in order to be published, the paper needs improvement with some aspects corrected and clarified.

Abstract

Line 13: why "microbial", if some BSC types contain lichens and mosses?

Lines 24-25: hereafter in the text – all genus and species names must be written in italic.

Introduction

Lines 49-52: What about the role of fungi in the stabilization of soil structure by mycelium?

Material and Methods

2.1. Site description, sampling and soil characterization – On my mind, these three parts should be described in separate subsections.

Lines 89-96: are there any references concerning the site characterization?

Lines 97-101: The description of biocrusts in the sampling sites is missing – e.g., their type (cyanobacterial, lichen-dominated, moss-dominated, mixed, etc.) and thickness.

Line 173: "Differences between the parameters" – what does it mean? The differences in parameters among sampling sites?

Line 178: "diversity indices of studied organisms" - incorrect expression, because diversity indexes are calculated for communities but not for organisms.

Line 181: Concerning PERMANOVA – for which purposes was the analysis performed and on which parameter was it based?

Lines 190-192: the expression is unclear.

Results

Line 211: "unique" – what does it mean?

Figure 1: Relative abundance on the vertical axis - of what? reads? OTUs?

Lines 247-248: "67% of fungal OTUs were identified by FUNGuild and only 22% of OTUs were assigned to different functional groups." What does it mean? What about the remaining 45% (67%-22%)?

Figure 2. I suggest the following caption: "Proportion of different functional groups in fungal communities of biocrusts at the western Iceland". And "100%" on the figure is not true because, as it was written above, "only 22% of OTUs were assigned to different functional groups."

Figure 3. The meaning of the letters on the bars should be explained.

Lines 323-324: "bacterial dissimilarities" - what does it mean? the dissimilarities in the composition of bacterial communities?

Discussion

Line 378: "corresponded to the sampling altitude" - what does it mean? the same altitudes of the Icelandic and Norway sampling sites?

Lines 378-382. It is not so clear why this information should be presented in such details in Discussion.

Lines 388-396. As a soil mycologist, I can definitely say that fungal communities in the studied crusts have been characterized very superficially. Such great part of the OTUs, which could not be assigned even to any phyla (in BOR – near 60%!!), looks alarming. Moreover, the conclusions on "ecophysiological traits for survival in cold environments" based on the analysis of mycobiota at the phylum level are problematic because such major phylum as Ascomycota contains great diversity of fungi with different and even opposite life-history strategies and ecological preferences. Additionally, the authors did not include into Discussion functional groups of fungi – it would be interesting even taking into account the fact that only small part of the OTUs were assigned to any functional group.

Lines 400-402: the sentence is unclear.

Conclusions

Line 462: "important implications" – what do these implications consist of?

In general, I have gotten the impression that such extremely broad spectrum of microbiota, which the authors tried to analyze in the studied BSC, influenced the quality of analysis decreasing its level.

All other corrections and suggestions are inserted into the PDF version of manuscript, which is attached.

Author Response

Reviewer 2.

The manuscript describes the microbial diversity in subarctic biological soil crusts at the western Iceland revealed by molecular approaches. The topic is ecologically interesting and important, but in order to be published, the paper needs improvement with some aspects corrected and clarified.

Abstract

Line 13: why "microbial", if some BSC types contain lichens and mosses?

Re: Modified into “communities of organisms”.

Lines 24-25: hereafter in the text – all genus and species names must be written in italic.

Re: Corrected throughout the manuscript.

Introduction

Lines 49-52: What about the role of fungi in the stabilization of soil structure by mycelium?

Re: The information about role of fungal mycelia was added to the text.

Material and Methods

2.1. Site description, sampling and soil characterization – On my mind, these three parts should be described in separate subsections.

Re: We separated this part into two sections: 1. Site description and 2. Sampling and soil characterization

Lines 89-96: are there any references concerning the site characterization?

Re: The climate data were recorded at the next climate station Borgarnes and this information was added to the text. Additionally, soil chemistry was measured in our previous paper, which has already cited in the manuscript (Pushkareva et al. 2021).

Lines 97-101: The description of biocrusts in the sampling sites is missing – e.g., their type (cyanobacterial, lichen-dominated, moss-dominated, mixed, etc.) and thickness.

Re: We added the information that mosses and lichens were not observed in the biocrusts. They were dominated by eukaryotic microalgae.

Line 173: "Differences between the parameters" – what does it mean? The differences in parameters among sampling sites?

Re: The sentence was modified as suggested.

Line 178: "diversity indices of studied organisms" - incorrect expression, because diversity indexes are calculated for communities but not for organisms.

Re: “of studied organisms” was deleted from the text.

Line 181: Concerning PERMANOVA – for which purposes was the analysis performed and on which parameter was it based?

Re: PERMANOVA analysis was performed based on amplicon sequencing dataset (number of reads). Using this multivariate non-parametric test we compared the sites and complemented the NMDS analysis. The information of used parameters was added to the text.

Lines 190-192: the expression is unclear.

Re: One sentence was deleted for better understanding.

Results

Line 211: "unique" – what does it mean?

Re: “Unique” was deleted from the sentence.

Figure 1: Relative abundance on the vertical axis - of what? reads? OTUs?

Re: The information was added to the figure title. “based on reads number obtained by amplicon sequencing

Lines 247-248: "67% of fungal OTUs were identified by FUNGuild and only 22% of OTUs were assigned to different functional groups." What does it mean? What about the remaining 45% (67%-22%)?

Re: This FUNGuild database is not fully complete, resulting in only partial identification of OTUs (67% in our case). Then, only part (22% ) of these 67% of identified OTUs could be assigned to the functional group.

This sentence was modified for a better understanding.

Figure 2. I suggest the following caption: "Proportion of different functional groups in fungal communities of biocrusts at the western Iceland". And "100%" on the figure is not true because, as it was written above, "only 22% of OTUs were assigned to different functional groups."

Re: The figure caption was modified as suggested. “Proportion of different functional groups of biocrust fungi identified by FUNGuild database”. The proportion is taken only for those 22% identified and assigned to functional group. This now is written in the caption.

Figure 3. The meaning of the letters on the bars should be explained.

Re: The sequences of isolates were submitted to the GenBank and these codes correspond to those displayed in there. Therefore, we believe that this column could be useful for some readers.

Lines 323-324: "bacterial dissimilarities" - what does it mean? the dissimilarities in the composition of bacterial communities?

Re: The sentence was modified as suggested.

Discussion

Line 378: "corresponded to the sampling altitude" - what does it mean? the same altitudes of the Icelandic and Norway sampling sites?

Re: The sentence was modified for better understanding. “corresponded to the altitudes of the Icelandic and Svalbard sampling sites”

Lines 378-382. It is not so clear why this information should be presented in such details in Discussion.

Re: we prefer to keep this information since it indicates that these cyanobacterial strains from Iceland are closely related to those reported on Svalbard. We added an addition statement: “Such data reflect a high degree of genetic relatedness and hence it is possible to speculate that, for example, migrating birds such as Arctic terns (Sterna paradisea) could act as vector for transportation of cyanobacteria between Iceland and Svalbard.”

Lines 388-396. As a soil mycologist, I can definitely say that fungal communities in the studied crusts have been characterized very superficially. Such great part of the OTUs, which could not be assigned even to any phyla (in BOR – near 60%!!), looks alarming. Moreover, the conclusions on "ecophysiological traits for survival in cold environments" based on the analysis of mycobiota at the phylum level are problematic because such major phylum as Ascomycota contains great diversity of fungi with different and even opposite life-history strategies and ecological preferences. Additionally, the authors did not include into Discussion functional groups of fungi – it would be interesting even taking into account the fact that only small part of the OTUs were assigned to any functional group.

Re: The discussion about functional groups was added to the text as suggested.

Lines 400-402: the sentence is unclear.

Re: The sentence was modified for a better understanding.

Conclusions

Line 462: "important implications" – what do these implications consist of?

Re: we deleted “important implications”

In general, I have gotten the impression that such extremely broad spectrum of microbiota, which the authors tried to analyze in the studied BSC, influenced the quality of analysis decreasing its level.

Re: Thank you for your opinion, we will consider it for our next studies.

All other corrections and suggestions are inserted into the PDF version of manuscript, which is attached.

Re: The corrections were made in the manuscript accordingly.

Round 2

Reviewer 2 Report

The manuscript looks better, but there are still a number of points that need to be clarified and improved.

Introduction

"Moreover, the mycelia of mycorrhizal fungi produce glycoprotein, which binds soil particles, and, thus, contribute to the soil stabilization". It is unclear why the authors cited namely this statement from Warren et al. (2019) – certainly, the Icelandic biocrusts do not contain sufficient amount of the mycorrhizal fungi (if any). In the cited paper, there is a much more appropriate statement: ”In general, fungi produce vegetative filaments (hyphae) which bind soil particles together and help to consolidate the surface of BSC communities".

Material and Methods, Site description

Once again - Are there any references from which the climatic, vegetation, and soil type data were obtained? – definitely, should be.

Sampling and soil characterization

"mosses and lichens were not observed in the samples." Does it mean that the Icelandic biocrusts belong to the cyanobacterial type?

Results

Table 1. Why are the column captions in a. and b. different?

3.2. Alpha and beta diversities.

Is OTUs richness the index?

Can 157 m a.s.l be considered a high altitude?

Why the data on community structure (composition), both taxonomic and functional, are not given in the subsection 3.1? On my mind, it is more logical to place these data on the communities' composition in different localities after the general data on communities' composition.

Concerning the Pearson correlation in Shannon diversity index among four microbial groups - for which purposes was this test performed if its results even have not been discussed in Discussion?

Discussion

Once again, the analysis of fungal communities in the studied biocrusts is superficial. On my mind, such great proportion of the OTUs, which could not be assigned even to any phyla, cannot be explained only by great amount of undescribed fungal species in Icelandic biocrusts or by poor coverage of fungal sequences in the existing databases. Yes, it is known that the UNITE database is the most popular database for the identification of fungal sequences, and that the ITS region is widely used as a barcode in fungal community studies, but it is also known that this region does not provide sufficient resolution for Penicillium spp. and Fusarium spp., which constituted the majority of isolated strains in the Icelandic biocrusts. Furthermore, Hibbett et al. (2016) listed several additional databases for the identification of fungal sequences. And what about fungal OTUs obtained by amplicon sequencing of the 18S rRNA, which constituted 10% of total eukaryotic reads and 58% of the Opisthokonta clade?

The last paragraph of 4.1. It looks more logical to discuss all algae including diatoms, and then – Cercozoa.

In Conclusion, the authors wrote about the "comprehensive inventory" of the biocrust microbiota. Once again, on my mind, its comprehensiveness is rather related to the extremely broad spectrum of microbiota studied, but not so to the quality of its analysis.

All other corrections and suggestions are inserted into the PDF version of manuscript, which is attached.

Author Response

Dear reviewer – many thanks for spending so much time with constructive comments to improve our paper, which is really appreciated. We tried our best to further revise the manuscript:

The manuscript looks better, but there are still a number of points that need to be clarified and improved.

Introduction

"Moreover, the mycelia of mycorrhizal fungi produce glycoprotein, which binds soil particles, and, thus, contribute to the soil stabilization". It is unclear why the authors cited namely this statement from Warren et al. (2019) – certainly, the Icelandic biocrusts do not contain sufficient amount of the mycorrhizal fungi (if any). In the cited paper, there is a much more appropriate statement: ”In general, fungi produce vegetative filaments (hyphae) which bind soil particles together and help to consolidate the surface of BSC communities".

Re: Corrected as suggested.

Material and Methods, 

Site description

Once again - Are there any references from which the climatic, vegetation, and soil type data were obtained? – definitely, should be.

Re: The climatic data were not published elsewhere and are not online available. We received them from local colleagues at the research station Litla-Skard. This information was added to the text.

“(meteorological data are provided from the research station Litla-Skard; no online data are available)”

Sampling and soil characterization

"mosses and lichens were not observed in the samples." Does it mean that the Icelandic biocrusts belong to the cyanobacterial type?

Re: Mosses and lichens were not observed in the collected samples. However, they are present in the area as it was mentioned in the site description. Moreover, in the sample description it is written that “Microphotoautotrophic communities were dominated by eukaryotic microalgae”, which indicate that the studied biocrusts were rather algal than cyanobacterial.

Results

Table 1. Why are the column captions in a. and b. different?

Re: Thank you for noticing this error. The caption of Table 1b was corrected.

3.2. Alpha and beta diversities.

Is OTUs richness the index?

Re: “OTUs richness” corresponds to Richness index (Observed OTUs) and it is a widely used term. If reviewer prefer, we could also change “alpha diversity indices” on “alpha diversity metrics”

Can 157 m a.s.l be considered a high altitude?

Re: We agree with the reviewer and changed “between low and high altitude” to “between the lowest and highest topographic elevations”.

Why the data on community structure (composition), both taxonomic and functional, are not given in the subsection 3.1? On my mind, it is more logical to place these data on the communities' composition in different localities after the general data on communities' composition.

Re: In our opinion, comparison of the sites rather belongs to alpha and beta diversities. We only use relative abundance dataset for showing the distinct features of site BOR, which also correspond to the alpha (richness and Shannon index) and beta (NMDS) diversity analyses. If reviewer insists on combining these two sections, we will change it accordingly.

Concerning the Pearson correlation in Shannon diversity index among four microbial groups - for which purposes was this test performed if its results even have not been discussed in Discussion?

Re: We agree with the reviewer and, thus, deleted this dataset from the manuscript.

Discussion

Once again, the analysis of fungal communities in the studied biocrusts is superficial. On my mind, such great proportion of the OTUs, which could not be assigned even to any phyla, cannot be explained only by great amount of undescribed fungal species in Icelandic biocrusts or by poor coverage of fungal sequences in the existing databases. Yes, it is known that the UNITE database is the most popular database for the identification of fungal sequences, and that the ITS region is widely used as a barcode in fungal community studies, but it is also known that this region does not provide sufficient resolution for Penicillium spp. and Fusarium spp., which constituted the majority of isolated strains in the Icelandic biocrusts. Furthermore, Hibbett et al. (2016) listed several additional databases for the identification of fungal sequences. And what about fungal OTUs obtained by amplicon sequencing of the 18S rRNA, which constituted 10% of total eukaryotic reads and 58% of the Opisthokonta clade?

Re: We added an additional sentence about limitations of UNITE database. Moreover, in the results section we added information on dominant fungal phyla from the 18S rRNA amplicon sequencing dataset.

The last paragraph of 4.1. It looks more logical to discuss all algae including diatoms, and then – Cercozoa.

Re: The paragraph was changed as suggested.

In Conclusion, the authors wrote about the "comprehensive inventory" of the biocrust microbiota. Once again, on my mind, its comprehensiveness is rather related to the extremely broad spectrum of microbiota studied, but not so to the quality of its analysis.

Re: This statement was deleted from the Conclusion.

All other corrections and suggestions are inserted into the PDF version of manuscript, which is attached.